# Endothelial Cell Participation in Inflammatory Reaction

**DOI:** 10.3390/ijms22126341

**Published:** 2021-06-13

**Authors:** Jean-Luc Wautier, Marie-Paule Wautier

**Affiliations:** Faculté de Médecine, Université Denis Diderot Paris, 75013 Paris, France; mpwautier@hotmail.com

**Keywords:** endothelial cells, inflammation, inflammasome, angiogenesis, thrombosis, SARS-CoV-2

## Abstract

Inflammation is an old concept that has started to be considered as an important factor in infection and chronic diseases. The role of leukocytes, the plasmatic components, then of the mediators such as prostaglandins, cytokines, and, in recent decades, of the endothelium has completed the concept of the inflammation process. The function of the endothelium appeared to be crucial as a regulator or the initiator of the inflammatory process. Culture of human endothelial cells and experimental systems made it possible to define the molecular basis of inflammation in vascular diseases, in diabetes mellitus, atherosclerosis, vasculitis and thromboembolic complications. Advanced glycation end product receptor (RAGE), present on endothelial cells (ECs) and monocytes, participates in the activation of these cells in inflammatory conditions. Inflammasome is a cytosolic multiprotein that controls the response to diverse microorganisms. It is positively regulated by stimulator of interferon response CGAMP interactor-1 (STING1). Angiogenesis and thrombotic events are dysregulated during inflammation. ECs appear to be a protector, but also a possible initiator of thrombosis.

## 1. Introduction

Blood circulation was described by William Harvey in the 17th century (1628 *Exercitatio Anatomica de Motu Cordis et Sanguinis in Animalibus*). Using the microscope, Marcello Malpighi carried out important work on capillaries and microcirculation. This led to his discovery of capillaries in 1661, which was fundamental to our understanding of the vascular system in the brain (1661 *De pumonibus epistolae II and Borellium*). The study of the vessel functions and pathology started to significantly progress in the 19th century. First in Europe—in Germany with Rudolf Virchow [1], in France Jean Cruveilhier [2], in England with William Osler [3]—and then in North America and in the whole world.

During the same period of time, physiologists started to understand the functions of the cardiovascular system. A new era was opened in 1970 with the culture of vascular cells. Animal endothelial cells and smooth muscle cells were first accessible and cultured in vitro. Additionally, vessel fragments were maintained in a survival state in bath and the functions were studied. Robert Furchgott, Louis Ignarro and Ferid Murad received the Nobel Prize in Physiology or Medicine (1998) for their discovery of the function of nitric oxide (NO) as a major regulator of vascular tone [4]. John Vane was distinguished with the Nobel prize in 1982 for his discovery of prostacyclin (PGI2) [5]. The reticuloendothelial system, associated with mononuclear system, was originally described at the beginning of the 20th century as a system that clears old cells and foreign bodies. This concept changed to the mononuclear phagocyte system.

During the 1980s, it was described that specialized endothelial cells (scavenger endothelial cells) phagocyted cell debris and dead cells. In the liver, sinusoidal endothelial cells, but not macrophages (Kupffer cells), were part of the system described by Ludwig Aschoff (Aschoff L. Das reticulo-endothelial system. *Ergeb Inn Med Kinderheilkd*, 1924).

The development of the culture of human endothelial cells was an essential step for understanding the physiopathology of vascular disease and inflammatory processes [6,7]. The researchers working on endothelial cells organized meetings in which they shared their results. This informal group became the association “Vascular Biology” during the meeting in San Diego (CA, USA) in 1992. Later, several scientists and medical doctors joined this group.

## 2. Endothelial Cells

Endothelial cells (ECs) are the most efficient antithrombic interface, and participate in the immune response to micro bacterial organisms. The predominant surface of exchange, which is the endothelium monolayer, is not a passive exchange barrier but a selective and adaptative frontier. Beside their barrier function, ECs have a determinant role in the thrombotic hemostatic equilibrium, first by heparinoid surface molecule, and then as receptors for coagulation components, von Willebrand factor (VWF), and Factor VIII coagulation factors. A fraction of plasma antithrombin is bonded to the heparan present on endothelial cells. Antithrombin activity is consequently located at the blood surface interface. The heparan sulfate insures the non-thrombotic properties of EC [8].

The functions of EC are difficult to explore in vivo since it is a unique linear cell inside the vessel. Fortunately, human ingenuity has made available endothelial cells in culture and facilitated the investigation of their functions.

### 2.1. Endothelial Cells In Vitro

#### 2.1.1. Human Umbilical Vein Endothelial Cells (HUVEC)

The method described by Maruyama [9] using human umbilical vein as a source of endothelial cells was subsequently developed by several research groups. The cord is separated from the placenta several hours after birth, placed in a sterile container, as described by EA Jaffe [7]. Human umbilical vein endothelial cells (HUVEC) are harvested from human umbilical vein after collagenase digestion. HUVEC are cultured in a specific medium. The cells reach confluence after 8–10 days [10].

Weibel-Palade bodies are storage granules present in endothelial cells. They contain P-selectin and von Willebrand factor, which is essential for hemostasis. P-selectin is an adhesion molecule participating in vascular permeability, and plays a role in leukocyte adhesion and platelet aggregation. Weibel-Palade bodies contain other molecules such as interleukine-8, endothelin-1, eotaxin-3, angiopoietin-2 and fructosyltransferase. These components are involved in inflammation, immune response, and angiogenesis [11,12,13] (Figure 1).

#### 2.1.2. Immortalized Human Endothelial Cells

Several attempts have been made to establish a continuous human endothelial cell line [14,15]. Simien Virus 40 (SV40) transfection has been proposed by several authors as a possible way of obtaining a permanent endothelial cell line [16,17]. An immortalized vascular endothelial cell line (IVEC) was obtained from human umbilical vein endothelial cells, which were transformed by intranuclear microinjection of a recombinant gene composed of the large T-encoding DNA of SV40 driven by the truncated vimentin promoter (HuVim 830-T/t). This immortalized vascular endothelial cell line shared most of the features of the original endothelial cells [18]. IVEC is a reliable and convenient cell line for studying endothelial cell reactivity in the presence of various stimuli. We studied IVEC proliferation and the expression of adhesion molecules induced by various cytokines. IVEC had a reduced sensitivity to IL-1β related to a decreased expression of receptor IL-1β, which can be corrected by dexamethasone treatment [19].

Since we observed unexpected results with two different cell lines of Human Microvascular Endothelial Cells (HMEC), as reported by Xu Y. [20], we tested these two types of human microvascular endothelial cells (HMEC-1, HMVEC-d) in parallel. The spontaneous proliferation of HMEC-1 measured by [H^3^-methyl] thymidine incorporation was higher compared to HUVEC.

#### 2.1.3. Arteriovenous Malformation-Derived Endothelial Cells (AMEC)

Endothelial cells obtained from arteriovenous malformations called AMEC [21] in culture demonstrated a higher propensity to proliferate and a reduced reactivity to antiproliferative compounds such as Interferon γ.

## 3. Vascular Permeability and Endothelial Cell Junctions

In the vessels, cell–cell junctions connect and modulate the activities of the individual endothelial cells, specifically for the control of vascular permeability and immune cell trafficking [22]. The cell–cell junctions in the endothelium are defined as tight junctions and adherens junctions, which alternate. Tight junctions strictly control vessel permeability at the blood–brain barrier [23]. The binding of vascular endothelial cadherin (VE-cadherin) to the same type of molecule is essential for the organization of the vascular network. GTPase-activating proteins (GAP) are particularly important, and define the local cortical versus radial organization of the actin fibers, thus regulating the plasticity of the junctions. VE-cadherin can also form complexes with signaling systems, like the endothelial specific VEGF receptor-2 (VEGFR-2), the vascular endothelial phosphotyrosine phosphatase, the transforming growth factor β receptor complex, and also the fibroblast growth factor receptor-1. VE-cadherin modulates signaling by limiting the nuclear translocation of catenins and other proteins that regulate cell transcription [24].

G-protein-coupled receptors can induce VEGFR-2. Gα13 is a member of the G protein family, which transduces signals from G-protein-coupled receptors through interactions with downstream effectors [25].

## 4. Endothelial Cell Receptors in Inflammation

Before the availability of endothelial cells (ECs) in culture, the interaction between blood cells and endothelial cells was investigated by means of transcapillary microscopy in animal or human eyes [26].

Observing the vessels in frogs, Ilya Mechnikov, called also Elie Metchnikoff [27,28], described phagocytosis, one of the main functions of leukocytes in the battle against infection. The characterization of adhesion molecules on endothelial cells provided insight into the pathophysiology of inflammation and atherosclerosis.

The adhesion of leukocytes is able to be significantly extended with the presence of inflammation, infection, and heart disease. Leukocyte adhesion is mediated by integrins β2 (CD18), α subunits (CD11a, CD11b, CD11c, CD11d) on leukocytes, Inter Cellular Adhesion Molecules (ICAM) and Vascular Cell Adhesion Molecules (VCAM) on the endothelium. This first step of leukocyte adhesion is then followed by interactions between members of the CD18 family and their respective ligands, which mediate firm adhesion/arrest of leukocytes on ECs. The leukocyte adhesion mediated by P-selectin can occur when inflammatory mediators such as histamine or thrombin activate ECs. The expression of E-selectin on the endothelial cell surface is indeed induced only by stimulation of ECs by inflammatory cytokines (e.g., IL-1, TNF-α). The intercellular cell adhesion molecules (ICAM) are also major adhesion molecules that arrest circulating leukocytes on ECs [29,30,31,32,33,34,35,36,37,38].

Monocytes and polymorphonuclear neutrophils in inflammatory conditions are associated with venous thrombosis. Monocytes secrete tumor necrosis factor α (TNF α). In vitro TNF α increased the shedding of Protein C inhibiting thrombomodulin production and stimulated tissue factor synthesis by endothelial cells and monocyte-macrophages (Figure 2).

Endotoxins, even at low concentrations, and tumor necrosis factor (TNF) activate endothelial cells, inducing ICAM and overexpression of VCAM. These endothelial cell receptors are involved in the recruitment of leukocytes. Activation of EC was associated with Interleukin-6 and Monocyte Chemoattractant Protein-1 (MCP-1) secretion. The coagulation was started by tissue factor production, which induced the cascade of the plasmatic factors, leading to the factor X coagulation complex (Tenase complex).

## 5. Physiology and Pathophysiology of Human Vascular Cells

### 5.1. Angiogenesis

Physiological angiogenesis is always transient, whereas pathological angiogenesis is uncontrolled. In experimental angiogenesis, growth factors act in an inflammatory context. Vascular endothelial growth factor (VEGF) and angiopoietin-2 are up-regulated, and angiopoietin-1 is down-regulated, which contributes to the loosening of matrix contacts and induces a switch from the quiescent phenotype to the activated one. The access of VEGF to VEGF-R2 VEGFR-1 induced a change in the activated phenotype to an angiogenic phenotype. The mechanism is incompletely understood and could be of major importance in diabetic retinopathy and new cancer angiogenesis. VEGF-induced endothelial stimulation and Platelet-Derived Growth Factor-BB (PDGF-BB) mediate pericyte recruitment, and co-expression of PDGF-BB normalizes aberrant angiogenesis, despite high VEGF doses. PDGF-BB co-expression anticipates the initiation of vascular enlargement and prevents VEGF-induced aberrant angiogenesis by modulating VEGF-R2 signaling, resulting in a limitation favoring vascular splitting into normal capillary networks [39]. Glycation of fibroblast growth factor 2 (FGF2) diminished its chemotactic effect toward endothelial cells. Glycated FGF2 displayed a lower angiogenic effect compared with no glycated FGF2. VEGF is a good candidate for triggering ocular neovascularization, since it is an autocrine factor for vascular endothelial and pigment epithelial retinal cells, its expression is up-regulated by hypoxia, it induces vascular permeability, and its glycation enhances its angiogenic activity (Figure 3). VEGF expression has also been found in several diseases in which angiogenesis is a prominent feature, such as rheumatoid arthritis, bullous pemphigoid, and psoriasis.

Angiogenesis and vasculogenesis, which occur during embryogenesis and body growth, are essential for organogenesis and tissue nutrition. Hemangiomas are different from vascular malformations. Hemangiomas are involutive tumors, whereas vascular malformations are composed of dysplastic vessels. Blood flow is extremely high in arteriovenous malformation. The tumor may alter the contiguous tissues. In vitro, the proliferation rate of arteriovenous malformation endothelial cells (AMEC) is not sensitive to the inhibitory activity of various cytokines (IL-1β, TNF-α, TGF- β, Interferon-γ) (Table 1). The low response to cytokines, the higher propensity to proliferate, and the ets-1 proto-oncogene expression suggest that AMEC have a defective regulation of proliferation that may be due a reduced apoptotic process [21].

Therapeutic angiogenesis has been explored as a strategy for restoring blood flow using VEGF-A. Clinical trials of VEGF gene delivery, despite its biological activity, have resulted in disappointing results. A better understanding of the factor regulation of angiogenesis is needed to improve this potent therapeutic approach.

### 5.2. Thrombotic Complications and Vascular Dysfunction

In several diseases, leukocytosis is a risk factor observed under inflammatory conditions, being an independent predictor of ischemic events. Inflammation decreases the natural anticoagulant mechanism, reducing protein C level activation and impairs the fibrinolytic system. Normal endothelial cell function is essential for maintaining vascular function. Increasing the risk of thrombosis alpha 2 antiplasmin is cross linked into the fibrin clot and inhibits plasmin; this leads to a reduction in fibrinolytic activity, favoring thrombosis. In addition to its vasodilatory effect, nitric oxide (NO) has a protective effect against vascular injury, inflammation, and leukocyte adhesion.

Von Willebrand factor (VWF) was discovered in people living in the Aland islands suffering from a hemorrhagic disease. It took decades to discover that VWF was a protein secreted by megakaryocytes and endothelial cells that makes a bridge between subendothelial structure and platelets. It was the initial step in stopping bleeding. VWF binds to the GPIbIX protein of platelets, followed by platelet aggregation, implying that the GPIIb IIIa platelet receptor is missing in Glanzmann Thrombasthenia Syndromes [46]. VWF was first thought to be a monomer (250 kDa), but this efficient protein is a multimer from 500 kDa to 10 million [47]. VWF binds to subendothelial structures, including collagen type I-III. This process is important in hemostasis, but may be inappropriate in inflammation where VWF is elevated and factor VIII c favors thrombosis.

A lack of von Willebrand factor causes increased vascularization. The molecular mechanism associates VWF to αvβ3 integrin, and possibly to angiopoietin-2 and galectin-3 which is similar to AGER3, a scavenger receptor. Angiodysplasia has been observed in congenital VW disease and acquired von Willebrand syndrome [48]. We described that anti-factor VIII-VWF may bind to endothelial cells and have a possible deleterious effect [49]. In a patient with chronic lymphocytic leukemia, we observed that anti-factor VIII-VWF was associated with vascular angiodysplasia [50], as was observed in congenital von Willebrand disease. Anti-factor VIII antibodies are frequently found in patients with lupus erythematosus. Anti-factor VIII antibodies could also be directed against factor VIII procoagulant or part of von Willebrand factor.

Endothelial cell dysfunction is a broad concept that implies reduced production of nitric oxide (NO) and disequilibrium between relaxing and contracting factors that may participate in the pathophysiology of various cardiovascular diseases [51,52,53]. In inflammatory conditions and atherosclerosis monocytes transmigrate into the arterial wall, passing between the EC. During some infectious diseases or other vascular diseases, endothelial cells, instead of protecting against thrombosis, become prothrombotic and may produce tissue factor [54].

### 5.3. The Receptor for Advanced Glycation End Products (RAGE)

RAGE, which is the main cell-surface molecule involved in the toxicity of Advanced Glycation End products (AGE), plays a crucial role in inflammation. The first identified was AGE R1 or RAGE present on endothelial cells, lymphocytes, smooth muscle cells and neurons. RAGE, a member of the immunoglobulin (Ig) superfamily, contains three Ig-like domains—one variable (V) and two constants (C1 and C2)—in the extracellular part, a single transmembrane domain and one short cytosolic tail [55]. The RAGE gene is present on locus 6p21.3, next to the CHM class III protein family. RAGE can bind a wide range of endogenous molecules, including AGE, the high mobility group box-1 (HMGB-1), also called amphoterin-c, β-amyloid peptide, and S100 calgranulins (Figure 4).

The binding of AGE to cell receptor RAGE induces a series of reactions, leading to reactive oxygen intermediates (ROI), intracellular activation, and stimulation of gene expression. Following AGE binding to RAGE, a series of reactions occur, resulting in NF-kB activation, gene transcription of inflammatory mediators, and increases in vascular permeability. These reactions lead to the idea that the AGE–RAGE axis is responsible in diabetes mellitus and metabolic syndrome for vascular dysfunction and glomerular sclerosis. Alongside this impact, AGE–RAGE seems to be involved in diabetic retinopathy and probably in Alzheimer’s syndrome.

First described as a cell receptor, RAGE is now acknowledged to generate several isoforms produced through alternative splicing or post-translational modifications. It has been reported that extensive splicing of RAGE transcripts leads to as many as 20 splice variants [56]. In endothelial cells, only three isoforms of RAGE have been detected at significant levels: N-truncated (Nt-RAGE), Full Length (FL-RAGE, usually called RAGE) and endogenous secretory (esRAGE). Other than by splicing, soluble RAGE (sRAGE) can also be produced, consequently leading to FL-RAGE proteolysis [57], and may act as a decoy, preventing RAGE engagement of ligands.

The distribution of these splice variants may differ according to the organ considered, and some of them are almost never synthesized. To date, the majority of these splice variants have not been detected as proteins. Soluble RAGE (sRAGE) found in human blood is the product of two constitutive mechanisms: alternative splicing (esRAGE) and proteolysis of RAGE. The cleavage of RAGE, upregulated by calcium, is achieved by many metalloproteinases, including ADAM10 and γ-secretase. It is still unclear whether the pathophysiological significances of esRAGE and sRAGE are distinct in different clinical situations, and by which organ or tissue sRAGE and esRAGE are produced. Many works have reported higher incidences of complications in patients with lower blood levels of sRAGE and/or esRAGE in diseases related or not to AGE: type 1 diabetes, type 2 diabetes, renal failure, atherosclerosis, metabolic syndrome, essential hypertension, longevity, Alzheimer’s disease, and vascular dementia. Serum levels of sRAGE are positively correlated with the inflammatory markers tumor necrosis factor-α (TNF-α) and Monocyte Chemoattractant Protein-1 (MCP-1) in type 2 diabetic patients, making sRAGE a potential biomarker of inflammation [58]

## 6. Inflammasome

The inflammasome is defined as a cytosolic multiprotein signaling complex that controls the response to diverse pathogen microorganisms [59,60,61].

It has been proposed that the activation of inflammasome not only stimulates the secretion of interleukin-1 (IL-1) family cytokines, it additionally induces the release of tissue factor (coagulation initiation) in activated macrophages. Inflammasome may be positively regulated by stimulator of interferon response CGAMP interactor-1 (STING1). Excessive activation of STING1 participates in the pathologic process of sepsis and initiates coagulation. STING1 is linked to inositol 1,4,5-triphosphate receptor type1, a calcium channel, and the ATPase sarcoplasmic/endoplasmic reticulum Ca^2+^ transporting 2 (ATP2A2 a calcium pump) mediates calcium influx. This calcium flux activates CASP1, CASP11 or CASP8 in macrophages/monocytes in response to different infections. STING1 depletion results in a reduction of coagulation activation. STING1 mRNA expression in blood mononuclear cells is correlated with the severity of disseminated intravascular coagulation in patients with sepsis. STING1 mediator type1 interferon response is of restricted influence for inflammasome-mediated coagulation [61].

### Inflammaging and Ageing

Cells are continuously communicating, functioning simultaneously as “sender and receiver”. This equilibrium varies depending upon the case (inflammation, oxidation, fibrosis, cancer), including during ageing. Inflammaging results from multiple origins, like proinflammatory tissue injury accumulation, and failure of the immune system to regulate and kill pathogens. A predisposition of senescent cells to produce proinflammatory cytokines, improved stimulation of the NFκB transcription factor, or the presence of a defective autophagic response. Senescent cells, in particular, are known to produce TNFα and IL-6 [62]. Inflammaging, a slow progressive inflammation, is additionally thought to be associated in ageing and associated with telomere alterations [63].

## 7. Inflammatory Conditions in Diseases

### 7.1. Atherosclerosis

In atheroma, macrophage foam cells can undergo apoptosis and contribute to vascular lesions and complications. Monocytes give rise to macrophages, exhibiting a proinflammatory program. Macrophages do not seem to be the sole leukocytes infiltrating the vessel, but they are predominant in the atheroma plaque. T and B lymphocytes are also present in atherosclerotic lesions. T lymphocytes produce interferon gamma, amplifying the expression of class II histocompatibility antigens. Regulating T cells secreting Transforming Growth Factor beta (TGF beta) may have a beneficial effect, limiting atherosclerotic evolution.

Antibodies secreted by B lymphocytes have a twin impact; the protect against infection agents like viruses, but may form immune complexes that have a toxic effect, activating the complement system.

### 7.2. Diabetes and Renal Failure

As a statistical risk, high glucose levels are a main factor for renal insufficiency associated with ageing. Historically, the definition of diabetes mellitus has varied, and has included glucose blood level, hemoglobin HbA1c level, glycosuria, and insulin blood level for type I diabetes. The medical definitions are a little less clear for metabolic syndrome, together with weight, hypertension, and renal function (glomerular filtration). The two metabolic disorders have several parameters in common, but the age of the patients is by far the most different. Unfortunately, no large well-conducted study has evaluated the risk for patients with metabolic syndrome in connection with various parameters. Insulin treatment improves the prognosis of type I diabetic patients. A blood glucose control of type II diabetic patients seems to be an efficient treatment with respect to morbidity and mortality. These totally different medical disorders have Advanced Glycation End Products (AGE), high blood and tissue concentration in common. AGE have more recently been identified as a risk factor. Previously an intermediate glycation component, HbA1c has been identified as a risk factor in diabetic patients. Structural proteins, including collagen, can be glycated, consequently modifying the mechanical properties. Plasmatic glycated proteins or cellular glycated components such as red cell membrane protein band 3 bind to receptors [64]. Several AGE receptors have been described, from RAGE, also named AGE R1, a multiligand receptor, to AGE R3, a scavenger receptor.

The possible pathophysiology results in the need to find an alternative treatment in order to avoid the deleterious effect. Reduced NO availability and reactive oxygen species formation are major disturbances of EC in diabetes mellitus. As has been known for a long time, diabetic control is a major factor, keeping blood glucose concentration within a normal range. Another aspect could be limiting or avoiding AGE formation and AGE binding to RAGE. First anti-RAGE antibodies were studied, then compounds blocking RAGE from avoiding binding to AGE were found, and their efficacy has been partially demonstrated in Alzheimer’s disease. As an alternative, the modulation of RAGE by drugs already known to be protective for vascular diseases has been proposed. Different studies conducted in Asia have tested the impact of traditional medicines and observed some beneficial effects. As research progresses, new advances appear. In vivo, in diabetic rats, blockade of glycated proteins binding to RAGE prevented increase in vascular permeability and oxidant stress. Infusion of recombinant soluble RAGE in hyperlipidemic diabetic rats prevented the development of accelerated atherosclerosis [65,66].

The balance between cellular RAGE (e-RAGE) and soluble RAGE (s-RAGE) resulting from the proteolytic action of ADAM 10 on cellular RAGE or genetically produced RAGE (r-RAGE) may be important for protecting against the deleterious effect of RAGE on the vascular system. Several studies have demonstrated that a low level of s-RAGE is a risk factor. These observations have led to the discovery of treatments that affect the equilibrium between s-RAGE and e-RAGE or en-RAGE. Clinical trials are being carried out, and are on the way to demonstrating their efficacy. Angiotensin Receptor Blockers (ARBs) are widely used to treat patients with hypertension, heart failure and diabetic nephropathy. The principal action of ARBs is to inhibit the interaction of angiotensin II (Ang II) and the Ang II type 1 (AT1)-receptor. Even though most effects of ARBs are attributed to blockade of the AT1-receptor, it is now obvious that members of the ARB class also possess intrinsic properties that are unrelated to the blockade of the AT1-receptor. ARBs have a wide range of action affecting metabolic pathways related to prostaglandins, thromboxane A2, nitric oxide (NO), Peroxisome Proliferator-Activated Receptors (PPARs) or oxidative stress. The effect of such agents on the reduction of oxidative stress could correct the endothelial alterations occurring during an inflammatory state and affect RAGE expression. Thus, we tested the effect of three ARBs, candesartan cilexetil, irbesartan and telmisartan on the expression of the mRNAs of RAGE isoforms in endothelial cells in culture. ARBs inhibited expression of RAGE isoforms and have a potential beneficial effect in RAGE modulation therapies [58].

### 7.3. Infectious Diseases and Vessel Alterations

Various infection agents have been suspected to alter vessels. Streptococcus infection was one of the first to be associated with cardiovascular complications. The heart valve alterations were observed in patients with acute rheumatoid arthritis. Vasculitis was detected in children with bacterial or viral infections, sometimes in clusters of patients with thrombotic thrombocytopenic purpuras [67,68].

Tuberculosis bacillus was suspected to be responsible of Kawasaki syndrome in several cases, but the pathogenesis is not very clear aside from the inflammatory state and the high interferon blood level. More recently, SARS-CoV-2 has been demonstrated to induce major vascular complications on multiple organs, including the lungs, heart, brain, kidneys, and vasculature. It has been proposed that SARS-CoV-2 could be considered to be an endothelial disease [69].

The virus binds to the Angiotensin converting enzyme (ACE) receptor and penetrates into vascular cells, leading to pulmonary vessel complications in a large number of patients. Less frequently, cardiovascular complications have been observed, but the binding of the virus seems to take place in the same way, entering coronary endothelial cells by the ACE receptor.

Markers of inflammation, such as C reactive protein, are augmented during SARS-CoV-2 pulmonary complications, as is also the case during acute myocardial infarction. The association of high CRP and elevated D-dimers is linked to a higher risk of death in patients with SARS-CoV-2. Patients with SARS-CoV-2 disease develop manifestations of shock. Proinflammatory cytokines such as TNFα, IL-1β, IL6, Granulocyte Colony Stimulating Factor (GCSF), Interferon γ and monocyte chemoattractant protein1 are significantly increased in patients with SARS-CoV-2. A high percentage (71.4%) of patients who have died showed evidence of disseminated intravascular coagulation.

The recent experience of SAR-CoV-2 pandemic infection has illustrated that inflammation and thrombosis are linked. Additionally, treatment should involve anti-inflammatory drugs such as anti-cytokine antibodies, corticosteroids and anticoagulants alongside antiviral drugs. This recent experience has suggested that EC are one of the main targets of the SARS-CoV-2 virus via the endothelial cell ACE receptor. The ACE receptor modulates RAGE, activating several intracellular pathways, and this could be the case in patients with hypertension or diabetes mellitus.

## 8. Conclusions

A paradigm in which inflammation involves endothelial cells has been proposed in the pathophysiology of atheroma, but has also more recently been proposed to be one of the major processes responsible for the severity of vascular lesions in SARS-CoV-2. The endothelial cell culture techniques we described make it possible to further understand the mechanism involved in blood cell–endothelial cell interactions in the inflammatory reaction occurring in acute thrombotic processes, as observed in myocardial infarction, thromboembolic complications, and disequilibrium between thrombotic and anti-thrombotic factors.

The vascular tree is not just a succession of tubes limited by the endothelium as a passive membrane. Endothelial cells represent a dynamic interface that has major functions in the homeostasis of body fluids. They are also a potential target of deleterious agents, lipids, AGE, bacterial endotoxins, and microbacterial agents. The modifications of endothelial cell functions lead to inflammation in different organs and tissue.

The major error we made was to consider that inflammation was unique. Inflammation is like a fire, and the way to combat it depends on the type of fire. Firemen know that wood, oil, petrol, and electrical fires have to be extinguished using different techniques. Unfortunately, in medicine, we are not at that level. 

Many nice and complicated diagrams have been drawn to explain how inflammation starts and progresses. No specific blockers or association of monoclonal antibodies has demonstrated superiority over previous treatment. Why? Are the concepts wrong? Certainly not. Are the clinical trials too expansive to reach to a conclusive protocol?

A new task force should be set up to better delineate what inflammation is and what treatment could be applied.

## Figures and Tables

**Figure 1 ijms-22-06341-f001:**
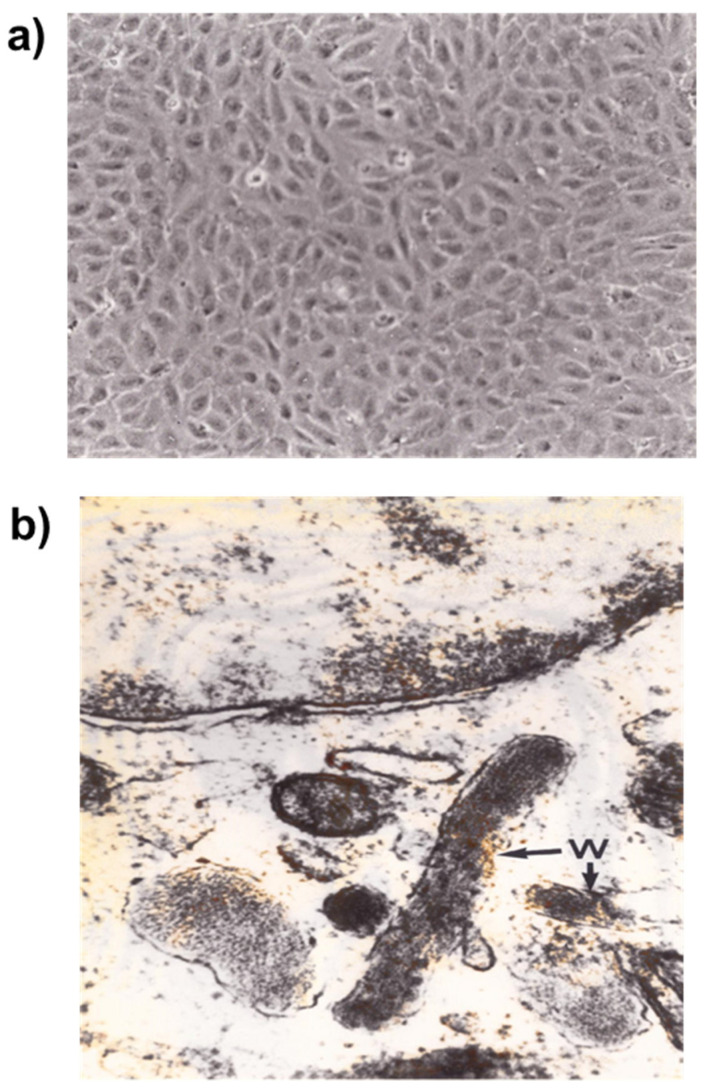
Cultured Human endothelial cells. Human Umbilical Vein Endothelial Cells (HUVEC) were cultured in Petri dishes until confluence and observed using an inverted microscope (×70) (**a**). In the ultrastructure the Weibel-Palade bodies (W) (electron microscope, ×50,000) used as a hallmark of endothelial cells, are secretory organelles used for post-synthesis storage in endothelial cells (**b**).

**Figure 2 ijms-22-06341-f002:**
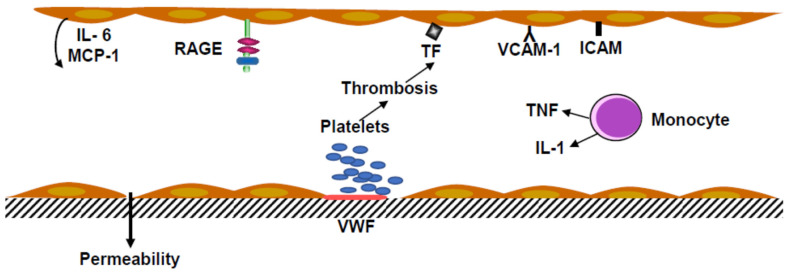
Inflammation and thrombosis in the vascular endothelium. Platelets adhering to the subendothelium, including von Willebrand factor (VWF) recruiting circulating platelets to form an aggregate. Leukocyte integrins bind to Vascular Cell Adhesion Molecule-1 (VCAM-1) and Intercellular Adhesion Molecule (ICAM). Activation of the receptor RAGE, a multiligand receptor, induces the release of Macrophage Chemoattractant Protein-1 (MCP-1) and Interleukin-6 (IL-6), and the production of Tissue Factor (TF). Monocytes stimulated by bacteria, endotoxins and viruses release tumor necrosis factor (TNF) and interleukin-1 (IL-1).

**Figure 3 ijms-22-06341-f003:**
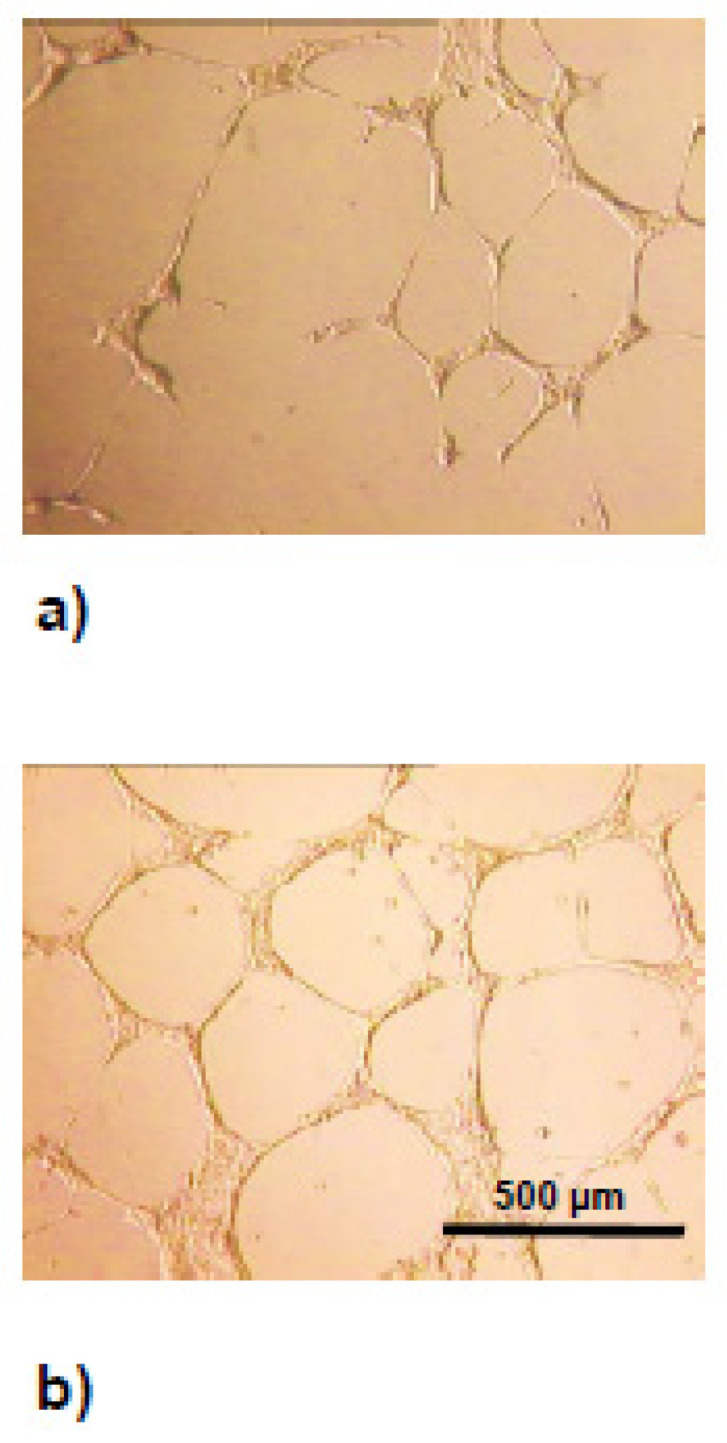
In vitro capillary tube formation. HUVEC grown on a synthetic matrix formed capillary tubes (11.2 ± 0.9 per mm^2^) (**a**). When HUVEC were cocultured with human peritoneal mesothelial cells (HPMC) activated by AGE or cytokines, they released Vascular Endothelial Growth Factor (VEGF), which promotes capillary tube formation (19.7 ± 0.6 per mm^2^) (**b**).

**Figure 4 ijms-22-06341-f004:**
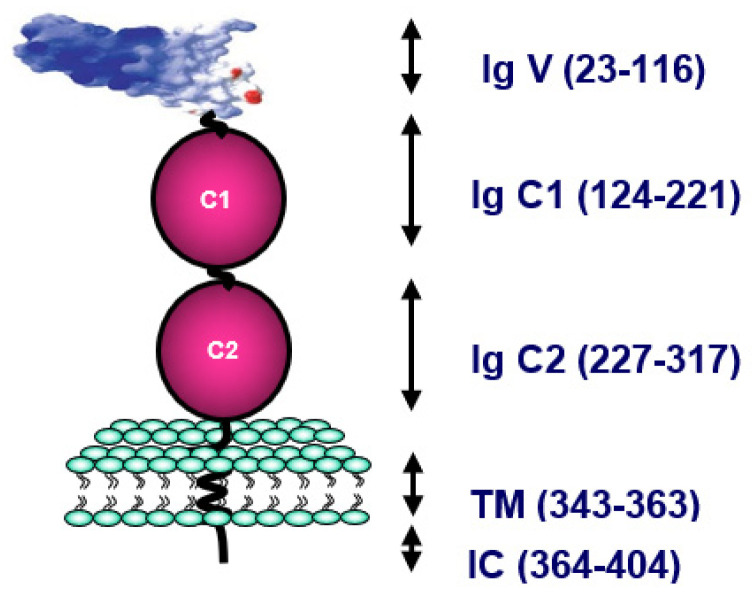
Molecular structure of the Receptor for Advanced Glycation End products (RAGE). Structure of full-length RAGE, including three Ig-like domains: the variable (Ig V) domain, two constants (Ig C1 and Ig C2) in the extracellular part, a single transmembrane domain (TM), and one short cytosolic tail (IC).

**Table 1 ijms-22-06341-t001:** Regulation of proliferation and survival of endothelial cells [40,41,42,43,44,45].

Positive Regulators	Negative Regulators
Vascular endothelial growth factor (VEGF))	Interferon
Platelet Derived Growth Factor (PDGF)	Interleukins (IL)
Fibroblast growth factor (FGF)	Tissue inhibitor metalloproteinase (TIMP)
Epidermal growth factor (EGF)	Angiostatin
Transforming growth factor (TGF)	Endostatin
Matrix metalloproteinases (MMP)	Plasminogen activator inhibitor-1 (PAI-1)
Tumor necrosis factor (TNF)	Thrombospondin (TSP)
Angiopoietins	Monocyte-derived endothelial cell inhibitory factor (MECIF)
	Endothelial-monocyte activating polypeptide II (EMAP II)

## Data Availability

It is a review article based on already published articles accessible on Pubmed, Google Scholar, Publons MDPI.

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
