# Peer review of "Endothelial Cell Participation in Inflammatory Reaction"

_ijms, 2021, doi:10.3390/ijms22126341_

Round 1

Reviewer 1 Report

The authors critically resumed many roles of endothelial cells in inflammation, but the review sounds disorganized and devoid of fundamental References. Moreover, more figures and tables might better support the text. Main criticisms are mentioned below:

Abstract-I suggest to rewrite the text focusing on the aim of this review. It seems to me a list of important pathogenetic mechanisms without clear correlation.

Introduction-At line 23, the authors must insert “…by William Harvey and by Marcello Malpighi in the 17th century (1628 Exercitatio Anatomica de Motu Cordis et Sanguinis in Animalibus; 1666 De polypo cordis).  Insert relative References.

At line 41 insert “Kupffer” and not “Kupfer” cells.

Line 47, the Figure 1 is not appropriate for a review article, it is better to delete it. In a magazine or a Commemorative article on the history of the foundation of a Scientific Society or Association, in my opinion, it will be suitable. The authors might insert a drawing of endothelial cells and their main functions.

Please insert more References in 2 Endothelial cells subheading.

I suggest to resume and collect all major abbreviations used here (in alphabetic order)  in “a List of Abbreviations” to put at the end of the text before References.

Figure 2 b is not explained in the text. Insert some sentences on the discovery and role of the Weibel-Palade bodies. The discovery of subcellular structures using TEM deeply characterized George Palade (Nobel Prize winner in 1974). In a review article, it is important the culture and historic revisitation of seminal discoveries.

3-Endothelial cell junction subheading-The text is poor and it is not clear the role and difference between tight and adherens junctions in the endothelium.

Line 124 insert “Metchnikoff” instead of “Mechnikov”.

Please insert much more References in the text when fundamental role of adhesion of leukocytes is associated to infection and heart disease at line 129 and   line 135.

The same criticism is referred to 5.1 Angiogenesis subheading. Please insert more citations, the authors must remember that a review generally includes almost 70 or 80 References and their critical comment.

Table 1 must be rewritten inserting the proper Reference/s for each line.

Lines 249-250 the author might insert a Figure on the AGE-RAGE axis and its function.

Please check better for English text. Some sentences are obscure. For example, at line 332 “The deleterious effect of high glucose level was possibly linked to the formation of AGE. AGE beside the binding to structural proteins, collagen binds to receptors.” must be rewritten.

Line 391 please insert more REFERENCES.

Conclusions are focused on the probable relation between SARS-CoV-2 and thrombosis, and vascular complications. This is an interesting aspect of the pandemia but not enough to be insert in the Conclusion.

In my opinion, this is only a side of a wider role of endothelial activity in diseases.

Author Response

The article was largely revised and is now suitable for publication. We divided our revision in one general comment and specific response to reviewer 1.

We thank the reviewers to help us to improve the manuscript.

According to reviewer 1 recommendations several modifications have been made and are highlighted in yellow. Figure 1 has been deleted.

According to reviewer 3 comments table 1 has been restructured since it was modified during the proof processing.

We added a new figure which has been redrawn, inspired from one we published (“The receptor RAGE in vascular and cerebral dysfunctions”, edited by Jean-Luc Wautier, Cambridge Scholars Publishing). We have the copyright, being the authors of chapter one and editor of the book.

Additional references are mentioned.

The authors critically resumed many roles of endothelial cells in inflammation, but the review sounds disorganized and devoid of fundamental References. Moreover, more figures and tables might better support the text. Main criticisms are mentioned below:

Abstract-I suggest to rewrite the text focusing on the aim of this review. It seems to me a list of important pathogenetic mechanisms without clear correlation.

We have some problem to understand what is a correlation between mechanisms involved in pathophysiology. A correlation is usually calculated using statistical test. Other relationship can be calculated using Fuzzy logic.

Introduction-At line 23, the authors must insert “…by William Harvey and by Marcello Malpighi in the 17th century (1628 Exercitatio Anatomica de Motu Cordis et Sanguinis in Animalibus; 1666 De polypo cordis).  Insert relative References.

Marcello Malpighi just born when Harvey published his article in 1628. He collaborated with William Harvey but defended his own thesis in 1661 in “De pulmonibus epistolae II ad Borellium”.  One reference was added.  (Lines 24-27)

At line 41 insert “Kupffer” and not “Kupfer” cells.

Kupfer is replaced by Kupffer (Line 44)

Line 47, the Figure 1 is not appropriate for a review article, it is better to delete it. In a magazine or a Commemorative article on the history of the foundation of a Scientific Society or Association, in my opinion, it will be suitable. The authors might insert a drawing of endothelial cells and their main functions.

The figure 1 was omitted. The functions of endothelial cells are published in books, review articles, and it is very difficult to draw all the functions of endothelial cells in one figure since endothelial cells are different in arteries, veins, organs.

Please insert more References in 2 Endothelial cells subheading.

I suggest to resume and collect all major abbreviations used here (in alphabetic order)  in “a List of Abbreviations” to put at the end of the text before References.

A list of abbreviations has been added

Figure 2 b is not explained in the text. Insert some sentences on the discovery and role of the Weibel-Palade bodies. The discovery of subcellular structures using TEM deeply characterized George Palade (Nobel Prize winner in 1974). In a review article, it is important the culture and historic revisitation of seminal discoveries.

A new paragraph has been added (Lines 74-80)

3-Endothelial cell junction subheading-The text is poor and it is not clear the role and difference between tight and adherens junctions in the endothelium.

This subject has already been explained in details in previous articles

Line 124 insert “Metchnikoff” instead of “Mechnikov”.

About the spelling of Mechnikov the way the reviewer 1 wrote is a French version of the name (Elie Metchnikoff) According to the Nobel price list the way we wrote (Ilya Mechnikov) is correct.

Please insert much more References in the text when fundamental role of adhesion of leukocytes is associated to infection and heart disease at line 129 and   line 135.

References have been extended

The same criticism is referred to 5.1 Angiogenesis subheading. Please insert more citations, the authors must remember that a review generally includes almost 70 or 80 References and their critical comment.

Table 1 must be rewritten inserting the proper Reference/s for each line.

The table 1 was unfortunately split in the proof and it has been corrected.

Lines 249-250 the author might insert a Figure on the AGE-RAGE axis and its function.

A figure of RAGE has been included (figure 4)

Please check better for English text. Some sentences are obscure. For example, at line 332 “The deleterious effect of high glucose level was possibly linked to the formation of AGE. AGE beside the binding to structural proteins, collagen binds to receptors.” must be rewritten.

This part has been reformulated.

Line 391 please insert more REFERENCES.

The total number of references is now 68

Conclusions are focused on the probable relation between SARS-CoV-2 and thrombosis, and vascular complications. This is an interesting aspect of the pandemia but not enough to be insert in the Conclusion.

The reasons why we mentioned SARS-CoV-2 in our conclusion are 1) the recent events 2) the number of deaths in the world per year, SARS-CoV-2: 3.7 million, diabetes: 1.9 million.

In my opinion, this is only a side of a wider role of endothelial activity in diseases.

This is the opinion of the reviewer and we agree, it is difficult to summarize all the endothelial functions in an article. We contributed to a multi author book and edited a book on RAGE in some diseases “The receptor RAGE in vascular and cerebral dysfunctions”, edited by Jean-Luc Wautier, Cambridge Scholars Publishing.  We previously wrote articles on endothelial cells. Our objective in this article was to focus on EC in some inflammatory conditions.

Reviewer 2 Report

The paper is well written and it tries to summarize the knowledge in this field.

Author Response

We thank the reviewers to help us to improve the manuscript.

Reviewer 3 Report

In my opinion, the manuscript is well written and is interesting; I only have one comment. 

1) Table 1: As the table extends over 2 pages, the column titles should be repeated at the start of the next page, together with the title "Table 1, continued"

Author Response

We thank the reviewers to help us to improve the manuscript.

According to reviewer 3 comments table 1 has been restructured since it was modified during the proof processing.

The table 1 was unfortunately spilt in the proof processing and it has been corrected. 

Round 2

Reviewer 1 Report

The present review has been ameliorated even if some minor changes still remain before acceptance.

  1. At line 130 insert inside brakets "(called also Elie Metchikoff ) [28]" at 
  2. Insert [28] as a new Reference: Underhill D et al "Elie Metchnikoff (1845-1916): Celebrating 100 years of cellular immunology and beyond" Nature Rev Immunology16, 651-656, 2016
  3. At line 196 rewrite "In vitro" in Italic font "In vitro"
  4. At lines 236-239 rewrite the full sentence that is not understandable in this form
  5. At lines 364-365 the sentence must be rewritten because it is not understandable

Author Response

We followed the recommendations of reviewer 1. The modifications are highlighted in yellow.